# A Comprehensive Community-Based Prevalence Study on Nocturia in Hong Kong Male Adults

**DOI:** 10.3390/ijerph18179112

**Published:** 2021-08-29

**Authors:** John Wai-Man Yuen, Ivy Yuen-Ping Wong, Peter Ka-Fung Chiu, Jeremy Yuen-Chun Teoh, Chi-Kwok Chan, Chi-Hang Yee, Chi-Fai Ng

**Affiliations:** 1School of Nursing, The Hong Kong Polytechnic University, Hung Hom, Kowloon, Hong Kong, China; ivy.y.p.wong@polyu.edu.hk; 2S.H. Ho Urology Centre, Department of Surgery, The Chinese University of Hong Kong, Shatin, Hong Kong, China; peterchiu@surgery.cuhk.edu.hk (P.K.-F.C.); jeremyteoh@surgery.cuhk.edu.hk (J.Y.-C.T.); chanck@surgery.cuhk.edu.hk (C.-K.C.); yeechihang@surgery.cuhk.edu.hk (C.-H.Y.); ngcf@surgery.cuhk.edu.hk (C.-F.N.)

**Keywords:** nocturia, quality of life, lower urinary tract symptoms, male adults, NQoL, urinary frequency, bedtime urination

## Abstract

Background: Most prevalence surveys on nocturia have focused on older populations. This study aimed to measure the nocturia prevalence across the full spectrum of male adults living in Hong Kong, where severity and associated quality-of-life (QoL) were also explored. Methods: A cross-sectional population-based survey was conducted in men aged 18 or above using the ICIQ-NQoL Questionnaire. Results: With 1239 respondents at age ranged 18–99, the overall nocturia prevalences were found to be 63.0% (ranged 41.6–84.6% at different age groups) and 31.2% (ranged 13.0–56.3% at different age groups), for ≥1 and ≥2 bedtime voiding episodes, respectively. The chance of nocturia was dramatically increased at age 60 or above while both prevalence and voiding frequency were increased with advancing age. About 83% of the nocturia men experienced one to two voiding episodes per night, but many of them had self-rated their QoL poor or very poor and indicated moderate-to-high bothersome. Younger men at age 18–39 were found to have high prevalence as 41.6% and almost 30% of them rated poor or very poor QoL. Conclusions: Nocturia was not only affecting the older populations but also caused significant bothersome and negative impacts on QoL in younger males.

## 1. Introduction

Many senior males are living with suboptimal sleep cycle and Quality of Life (QoL) that could be caused by the problem of nocturia. Nocturia is one of the most common lower urinary tract symptoms (LUTS), which describes the condition of waking at night to void while each void is preceded and followed by sleep [1]. The condition of nocturia is aligned with the nonrestorative sleep aspect of insomnia, as given in the full definition that “insomnia is a patient-reported problem characterized by difficulty falling asleep or difficulty maintaining sleep; i.e., frequent awakenings, difficulty returning to sleep after awakenings, or awakening too early with inability to return to sleep” [2]. Individuals with nocturia were suffering from sleep disturbance, but also commonly experiencing decreased levels of well-being, general health, vitality and essential biological rhythms [3,4]. Sleep quality was identified as a major mediator for several QoL domains impacted by the nocturia [5]. The association between ageing and nocturia could be at least partially explained by their comorbidity with age-related conditions ranging from urological illnesses to metabolic and neurological disorders; however, increasing trends have also been observed in younger men [6,7,8,9]. From the sleep expertise perspective, the interplaying role of obstructive sleep apnea was proposed to explain the bidirectional mechanisms between nocturia and insomnia through the increases in arousals, intra-abdominal pressure, atrial natriuretic peptide secretion [10]. By hypothesizing nocturia as an important clinical indicator of obstructive sleep apnea with mounting evidence, the physiological and psychological aspects underlying the nocturia and sleep relationship could be correlated with several nocturia-associated conditions, namely benign prostate hyperplasia, overactive bladder, diabetes mellitus and depression [10]. Approximately half of the insomnia cases were found to have a chronic course implicated with different epidemiological and functional correlates that are also observed in nocturia [2]. Medications and lifestyle habits were identified as additional causes of nocturia affecting all age groups [9,11,12]. Nonetheless, the multifactorial nature of nocturia was shown to impact QoL negatively, which has drawn much public health attention. 

In general, the urology field believed that at least two wakes per bedtime sleep are needed for causing clinical significance to affect energy metabolism and productivity [13,14]. Prevalence of nocturia reported from numerous studies were referred as both ≥1 and ≥2 voids per night. By gathering all English studies from 1990 to 2009, the nocturia prevalence for ≥1 and ≥2 voids were reported, respectively, as 11–35% versus 2–17% among men at age 20–40 while 69–93% verses 28–62% among older men at age >70; varying across different geographical locations [15]. In Asia, LUTS affected 60% of men ≥40 years and with 38% were nocturia, which was regarded as the most prominent and bothersome urinary storage symptom [16]. In a study conducted among five Asian countries, older males at age 50–80 living in Hong Kong has ranked as the second highest prevalent population for moderate-to-severe LUTS at 48% just following the 59% prevalence of the Philippines [17]. In Hong Kong, nocturia was identified as the most prevalent LUTS affecting two-thirds of men ≥40 years who have reported two or more voids per night [18]. Recently, local prevalence of nocturia were reported exceptionally high as 93% and 77% for ≥1 and ≥2 voids per night, respectively in male patients surveyed in multiple public urology clinics, and many had self-reported that ‘some” or “a lot of” bothersome feeling was caused by nocturia [19]. However, another telephone-based survey has revealed a relatively lower prevalence among men aged >40 in the community at 63% for ≥1 void and 32% for ≥2 voids per night [20]. The local age-specific prevalence in adult males was only reported at selected older age groups, such as 64% for age 51–60 and 81% for age >70, for having ≥1 void per night [21]. Furthermore, in a small-scale local interview study, 60% of newly retired men had reported that their sleep was disturbed by nocturia in the past six months, and more than half of them had not taken any actions for solving this problem but felt distressed [22]. Yet, the previous prevalence studies were mainly focused on older populations of selected age groups either from clinical settings or through phone interviews [20,21]. Therefore, the current study was conducted to clarify the population-based prevalence of nocturia and measure its impacts on health-related QoL using the Nocturia Quality of Life module of the International Consultation on Incontinence Modular Questionnaire (ICIQ-NQoL), in adult males living in Hong Kong. 

## 2. Materials and Methods

### 2.1. Study Design & Subject Recruitment

This cross-sectional study adopted a street-intercept and random walk survey design to measure the prevalence of nocturia and its health-related QoL perceived by adult males living in Hong Kong, using the Cantonese version of the ICIQ-NQoL. The target population of this study was adult males who were of Chinese ethnicity at age ≥18 and staying in Hong Kong as their usual residential place for >180 days prior to the survey. Foreign domestic helpers and individuals who could not understand and speak Cantonese were excluded. According to the Hong Kong census [23], in the middle of 2016, there were 2,846,845 adult males living in the city, with 92.0% belonging to the Chinese ethnicity and 88.9% having Cantonese as the usual language who were eligible to participate in this study. During July to August 2016, data collectors were allocated to visit different public spots (mainly parks, recreation facilities, and transportation hubs) of residential areas in 13 districts for sampling. The accessibility covered 78.8% of the entire targeted population, as the commercial, industrial and border districts, rural areas and remote islands were excluded in this study [23]. Each data collector spent 3–4 h at each allocated spot at each visit, and 5–6 questionnaires were completed on average per hour. In case of non-response, the next individual would be approached, and call-backs were not implemented. The nature and purpose of the study was verbally explained after confirming the eligibility, then the individual was invited to complete the anonymous questionnaire. Because the survey was conducted in the public street, verbal consent was obtained from the participant for attempting the questionnaire, who has the right to withdraw from participation without any penalties. Ethical approval has been obtained from the Human Ethical Sub-committee of the Hong Kong Polytechnic University. 

### 2.2. The Instruments and Measurements

The survey questionnaire began with two questions on the episodes of bedtime urination (0, 1, 2, 3 or ≥4 episode per bedtime) and frequency of waketime urination (every hour, every 2 h, every 3 h or ≥every 4 h). Two additional questions were used to ask the respondents about how much the bother was caused by the bedtime and waketime urination frequencies. The ICIQ-NQoL questionnaire was developed and validated for measuring the impacts of nocturia on QoL [24]. The Hong Kong Cantonese version (Figure A1) was translated and back-translated from the original English version obtained from ICIQ webpage (http://www.iciq.net/ICIQ.nqolmodulepage.html (accessed on 25 August 2021)). The validation protocol suggested by the ICIQ group was followed. Bilingual native speakers of Hong Kong, being a Cantonese speaker and English speaker were invited to conduct the initial and back translation, respectively. The back translated version was reviewed by the ICIQ group, and appropriate adjustment was made as recommended. Finally, the translation was accepted following the completion of the pre-test with 18 adult males who have completed the translated questionnaire and a set of interviewer questions provided by the ICIQ group. Those 18 adult males were recruited from the Urology Clinic of the Prince of Wales Hospital, and the one-month test-retest reliability for the Cantonese ICIQ-NqoL was determined as good with an overall test-retest coefficient of 0.79 while 0.69 for factor-1 and 0.87 for factor-2, which were also acceptable. The Cronbach’s alpha reliability based on the current sampled population was found to be highly reliable with α = 0.90, 0.87 and 0.85 for the overall, factor 1 and factor 2 subscales, respectively.

Constructs of the original ICIQ-NQoL remain unchanged, which consists of 13 items to measure the overall and two factor scores. The scoring scheme was based on the 5-point Likert scale (from 0 to 4) of ‘frequency’ on having the feeling described in each item in the past two weeks. The overall NQoL score ranged 0–52 was calculated by summing up the scores of all 13 items, with greater value indicating higher impact on QoL. Whilst the factor scores of the subscales were calculated by summing up the scores of corresponding items, i.e., 6 items (No. 1, 2, 3, 4, 5, 7) in factor-1 score for measuring the impact on ‘sleep/energy’ ranged 0–24; and 7 items (No. 6, 7, 8, 9, 10, 11, 12) in factor-2 score for measuring the impact on bother/concern ranged 0–28 [24]. The item 7, ‘difficult to get enough sleep at night’, was overlapping in both subscales. 

Demographic variables included age, marriage status (single, married or cohabited), the highest education level attained (college or above, secondary, primary or none) by the respondents, smoking habit (current smoker, have quitted smoking or non-smoker), and the drinking habit (Daily drinker, social drinker or non-drinker). Health and illnesses variables included self-rated imaginable health state as measured by using the EuroQol visual analog scale (EQ-VAS) ranged from 0 (the worst) to 100 (the best) in Figure A2 [25], and categorical items on whether respondents were having any of the following known nocturia-associated illnesses: diabetes mellitus, hypertension, comorbidity of diabetes mellitus and hypertension, benign prostatic hyperplasia and use of medication for it, and prostate cancer. 

### 2.3. Date Processing and Statistical Analysis

For studying the impact on QoL and its characteristics, the definition of International Continence Society for nocturia was adopted, and therefore, the nocturia population of this study is defined as those who had ≥1 void per night. Data collected from the survey were entered and analyzed using the SPSS version 25.0 (IBM, Armonk, NY, USA) and the Prism version 8.0 (GraphPad, San Diego, CA, USA). The prevalence of nocturia was estimated according to the overall and age groups of 18–40, 41–50, 51–60, 61–70 and ≥71. Descriptive statistics were used for reporting the categorical variables (frequency, percentages and confidence interval 95%) and continuous variables (mean and standard deviation (SD)) of the demographics, health conditions, urination frequencies and bothersome. Linear correlation between two continuous variables were evaluated using the Pearson’s correlation analysis. Chi-squared (χ^2^) test, students’ *t*-test, and one-way ANOVA were used to assess the significant differences of nominal and continuous variables between two groups. The significance level was sought at two-tailed *p*-value of 0.05. 

## 3. Results

### 3.1. Prevalence and Characteristics of Nocturia

A total of 1239 men were recruited from 3672 male pedestrians who had been approached (with a response rate of 33.7%) in multiple districts of Hong Kong. As shown in Table 1, the studied population reported a mean age of 56.7 ± 15.2 (ranged 18–99), two-thirds were married, almost half were current smokers, around one-fourths were daily drinkers, and 60.0% received education below college level. 

In total, 781 out of the population reported at least one bedtime void per night, which estimated an overall prevalence rate of 63%, which was reduced to 31.2% when only those experiencing two voids per nights were counted. When compared with those without nocturia, the nocturia group perceived a poorer health status (70.2 ± 12.6 versus 76.8 ± 11.4; *p* < 0.001) with more encounters of various health conditions including diabetes mellitus (DM), hypertension (Hx), DM-Hx comorbidity, benign prostatic hyperplasia (BPH) and use of medications for treating BPH (Table 1). As is also shown in Table 1, the nocturia was associated with less educated (< 0.001) and smoking (*p* < 0.001).

In the studied population, the prevalence of nocturia was increased with advancing age and observed in exactly the same trends between ≥1 and ≥2 voiding frequencies (Figure 1). The upwards trends were segmented into two linear curves by age 60, with relatively steady gradual increases (41.6–54.4% and 13.0–21.8%) from age 18 to 60, and then followed by a rocketing increase to 79.2% and 44.2% at age 61–70 and to 84.6% and 56.3% at age ≥71, with voiding frequency ≥1 and ≥2, respectively (Figure 1). In particular, for younger adult males at age 40 or below, high prevalences were reported as 42% for ≥1 void and 13% for ≥2 voids (Figure 1). 

As shown in Table 2 and Table 3, the number of nocturia episodes was in clear linear correlation with the age (r = 0.329; *p* < 0.001). Despite the vast majority of nocturia population (83.0%) being reported with experience of one to two voiding episodes per night, as a sum of 50.5% and 32.5% for voiding once and twice, respectively (Figure 2a), the remaining 17% with three to four voiding frequency or more were remarkably older at age (69.9 ± 10.1 versus 58.6 ± 13.9; *p* < 0.001). Figure 2b indicated that voiding frequency ≥3 was rarely reported by individuals below age 40 at around 1.5%, but the prevalence was elevated dramatically to 4.6–9.9% at age range of 41–60 and 19.4–32.1% at age >60 (Figure 2c–e). Specifically, for those at age ≥71, high prevalence rates were found as 20.1% and 12.0% for three and four voiding frequencies, respectively (Figure 2f).

### 3.2. Nocturia-Related QoL and Bothersome 

The ICIQ-NQoL scale was used to measure the negative impact of nocturia on QoL. Overall, 14.9% of the nocturia males have rated their QoL level poor/very poor with 44.0% and 32.8% of them were having one and two voids per night, respectively (Table 4). This indicated that the QoL of individuals could also be significantly affected by just one to two nocturia frequencies, despite almost 30% of individuals experiencing ≥4 voids per night were reported poor/very poor QoL rating (Table 4). In particular, when comparing between individuals with nocturia who were above and below age 40, the percentages of respondents rating their overall QOL poor or very poor were 14.9% versus 29.2%, respectively.

The studied nocturia population reported a mean self-rated NQoL score at 14.1 ± 9.1 out of the total score of 52, with the sleep/energy (factor 1) and bother/concern (factor 2) scores were 6.0 ± 4.8 and 7.7 ± 5.4, respectively. The overall NQoL and its factor scores were shown to follow the same trends with the voiding frequency (Figure 3a) and age of the respondents (Figure 3b). As shown in Table 3, age of respondents was significantly correlated with both NQoL score (r = 0.103; *p* < 0.01) and factor 1 score (r = 0.137; *p* < 0.001) while nocturia frequency was significantly correlated with the overall and two factor score (r = 0.253–0.301; *p* < 0.001). The clear increasing trends observed in Figure 3a suggested voiding frequency as a severity indicator. On the other hand, increasing trends were observed with age above 40; however, the overall and factor scores of the age group at 18–40 were higher than those measured in the age group of 41–50 and at equivalent level with the 51–60 age group (Figure 3b). The item ‘difficult to get enough sleep at night’ was identified as the commonest QoL aspect that has factored both the ‘sleep/energy’ (factor 1) and ‘bother/concern’ (factor 2). For factor 1, difficulty in getting enough sleep at night has caused individuals to nap during the daytime, which was shown to affect at least one-fourth of individuals experiencing ≥1 void while the highest with over 60.0% of those experiencing three voids. The other two items ‘difficult to concentrate the next day’ and ‘feel no energy the next day’ were predominantly affecting nocturia male individuals experiencing ≥2 bedtime voids. For factor 2, ‘concerned that must get up in the middle of the night to urinate’ was the most frequent single aspect particularly concerned by more than half of individuals having ≥3 voids. Whilst another item ‘pay more attention of when and how much to drink’ has also been the concern of 30–50% of the nocturia males, whereas the observed trend was in a reverse relationship with increasing voiding frequency. 

Approximately 40% of nocturia men have rated a score of 5 or above to indicate moderate-to-high bothersome levels caused by the bedtime urination, despite the mean score of the whole population being as low as 3.7 ± 2.7 out of 10. Consistent with the NQoL, the bothersome level was also elevated with the bedtime voiding frequency, whereas the bothersome score was peaked at 6.0 among those experiencing ≥4 voids per night (Figure 3c). Besides, many nocturia males were also bothered by their waketime urination frequency. As compared with the non-nocturia group, waketime urination in every 1–2 h were about 20% more prevalent among nocturia males who also reported a significantly higher (*p* < 0.001) bothersome score (Table 1). Nonetheless, as shown in Figure 3c, the bothersome scores of both bedtime and waketime urinations have followed similar upward trends with increasing nocturia frequency. Furthermore, the voiding frequency and health status of nocturia males were in a negative correlation relationship, where the health status score was decreased from the highest of 71.4 ± 12.1 for those experiencing one void to the lowest of 66.2 ± 18.1 for ≥4 voids (Figure 3d).

## 4. Discussion

The present comprehensive prevalence study estimated that 63% of Hong Kong male adults were living with nocturia, whereas its occurrence was clearly advancing with age affecting anywhere between 42%–85% from individuals of different age groups. The age-specific prevalence across the whole age spectrum of male adults ranging 18–99 years were reported and in accordance with the voiding episodes. Results of the ICIQ-NQoL measurement revealed that the QoL of men experiencing nocturia was affected at low-to-moderate levels that caused significant bothersome levels. 

First of all, the current prevalence results were consistent with the report of a recent telephone-based survey conducted in Hong Kong [20]. These community-based prevalence levels were remarkedly lower than those reported in the local clinical settings [19]. Comparing with other Asian and Western countries [26,27,28,29], Hong Kong was shown to be at relatively high nocturia prevalence. It is agreed with the report amongst five Asian countries that Hong Kong was known to be the highest prevalent place for LUTS, where nocturia was dominant among elderly males [17]. Such high prevalence may be at least partially explained by the ageing population and stressful living environment emersed in the city. Current results also suggested that the chance of nocturia was dramatically increased at ages after 60, which was in line with the age-dependent nature of nocturia [29]. In general, accumulating evidence [7] has suggested the associations of nocturia with numerous age-related illnesses that were also identified in the current nocturia population. Besides ageing and related health conditions, two demographic factors were particularly shown to be strongly associated with nocturia. One was the smoking habits, whose associated risk for nocturia has been reported in the Chinese male population [29]. Whilst the lower educational level received by the nocturia males, as reported in this study, is believed to be identified for the first time as associating with the nocturia occurrence. A possible postulation could be that the lower social class, commonly less educated, may have longer working hours and be less able to urinate during work, so have a tendency for drinking less in the daytime but more after work (to refill) and late dinner, which results in nocturia. This population could also be possibly living with a low income and self-care skills, such as buying healthy food, taking care of one’s own body, and carrying out preventive tests. Furthermore, the present findings also revealed a relatively higher age-matched prevalence for ≥1 void in younger men under 40 years old, as compared with the 35% reported in the large European Prospective Investigation into Cancer and Nutrition (EPIC) cohort study conducted in major western countries [26]. Kalejaiye et al. [30] discussed the possible causes of LUTS and storage symptoms with predominance of nocturia. In relation to this, another research team [31] had investigated the urodynamic functions in 308 young men experiencing non-prostatic-related LUTS and found that 80% of them were presented with storage symptoms. Pathologically, it was well established that nocturia could be caused by the decreased bladder storage capacity or increased urine output during sleep [7,8,11]. The former signifies the nocturia in younger individuals with the latter predominantly occurring as nocturia polyuria (NP) in older adults as characterized by an increased proportion of the 24-h urine output at night [8,32]. The age-related voiding frequency could be more accurately determined by the interaction between the rate of urinary output and the reservoir capacity of the low urinary tract [7], which was impossible to be determined in the current community-based survey conducted in the streets. However, the etiologies of nocturia, especially for younger male populations, required further investigations.

Recent research has paid more attention to how QoL is impacted by nocturia. In the studied population, the NQoL scores measured by the ICIQ-NQOL scale, bothersome scores caused by bedtime urination and perceived health status scores were all correlated with the bedtime voiding frequency. The varying trend of health state scores against the bedtime voiding frequency (as shown in Figure 3d) suggests that the perceived health status might be influenced by multiple factors, which warrants more in-depth investigations into the characteristics associated with the nocturia episodes. Participants of this study were asked to rate the 13 ICIQ-NQoL items based on their experience due to the bedtime waking for urination during the past two weeks. The single item ‘difficult to get enough sleep at night’ was identified as the most frequently responded to by the participants with nocturia, which was commonly factored into ‘sleep/energy’ and ‘bother/concern’ subscales. This was consistent with the notion of numerous studies that nocturia causes sleep disturbance and subsequently affects daytime activities, productivity and social activities [12,13,33]. Individuals who need to wake for urination at night were commonly suffering from certain degrees of sleep disturbance, which resulted in decreased levels of well-being, health, vitality and essential biological rhythms [26]. Nocturia was known to affect QoL both physically and mentally, especially as it was associated with increased incidence of depression [34]. In accordance with the current and previous studies, there was no doubt that the negative impacts of nocturia on QoL were dependent on the advancing age and increasing frequency of bedtime voiding, as also identified in present study. Evidence has suggested that the energy and productivity of any individuals could be substantially influenced if nocturia occurs at twice or more voiding episodes per night [14]. On the contrary, this current study revealed that many of the respondents experiencing even a single bedtime void had self-rated a poor or very poor QOL with significant bothersome levels caused by nocturia, especially for those younger men who were under 40 of age. Nocturia and its characteristics are seldom studied in the younger populations. Considering that sleep disturbances are complained of by many adolescents and young adults [2,35], current results may reveal nocturia as a possible health concern hidden among this population. The bidirectional relationships between nocturia and sleep quality in young men are multifactorial and could be implicated by actual difficulty in falling asleep or simply some kinds of bad behaviors, such as staying up late for game playing, alcohol drinking, and partying [2]. Given the fact that a vast majority of individuals were experiencing bedtime voiding limited at one to two episodes per night while ≥3 voiding episodes per night were predominantly observed at the age above 60, nocturia was shown to affect the QoL of all adult male ages regardless of the urination frequency. More investigations are needed to understand more in-depth on how men’s QoL and other health-related parameters are influenced by nocturia, especially for those of a younger age. 

The current study has advantages on previous reports, as it has included the whole age spectrum of male adults, and the well validated ICIQ-NQoL scale was used for evaluating the health-related QoL associated with nocturia. The random walk sampling method used in this study is regarded as a cost-effective sampling approach for conducting a community-based survey without scarifying of generality [36]. The authors are aware of the following four limitations due to the study design: (1) A complete subject list is unavailable since participants were invited and surveyed in the public street, where personal information was avoided; (2) unable to recruit individuals who were staying indoors and did not go out to the streets to participate; (3) certain subgroups of the target population might have been missed out due to the date and time being chosen for conducting the survey, for example, people who were working in an office if it was within working hours; and (4) some very small side streets might be ignored by the data collectors. As compared with the target population size, this study incurred possible bias, considerably because of the small sample and response rate, where pedestrians being invited to participate may not feel comfortable to talk about their urination behaviors in an open area. However, the resultant overall prevalence estimated was consistent with previous similar surveys.

## 5. Conclusions

Nocturia was prevalent in over 60% of male adults living in Hong Kong. Although both prevalence and voiding frequency were increased with advancing age, it bothered all ages by causing sleep disturbances. Despite the chance of nocturia being shown to be dramatically increased at age 60 or above, high prevalence was observed in younger men who also reported poor QoL. In conclusion, nocturia not only affected the older populations but also caused significant bothersome and negative impacts on QoL in younger males.

## Figures and Tables

**Figure 1 ijerph-18-09112-f001:**
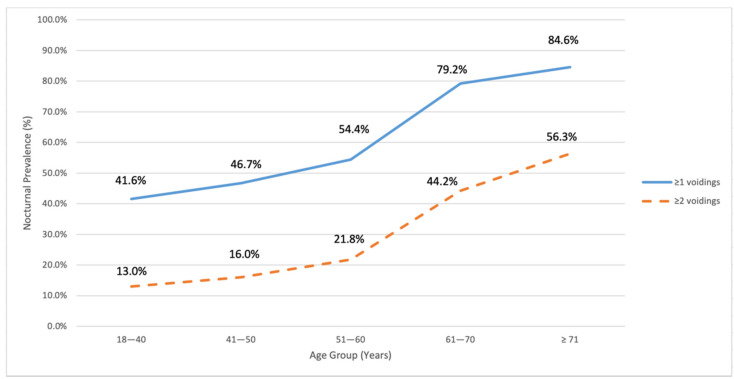
Prevalence of nocturia in different age groups.

**Figure 2 ijerph-18-09112-f002:**
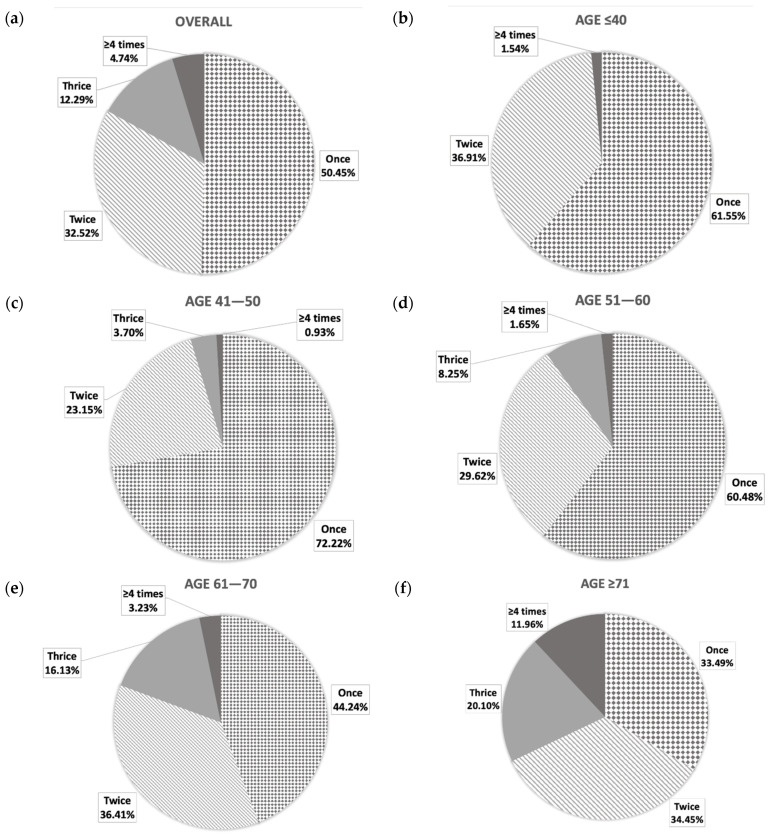
Distribution of nocturia episodes in: (**a**) the overall studied nocturia population, which was divided into different age groups at: (**b**) 18–40 years; (**c**) 41–50 years; (**d**) 51–60 years; (**e**) 61–70 years; and (**f**) ≥71 years.

**Figure 3 ijerph-18-09112-f003:**
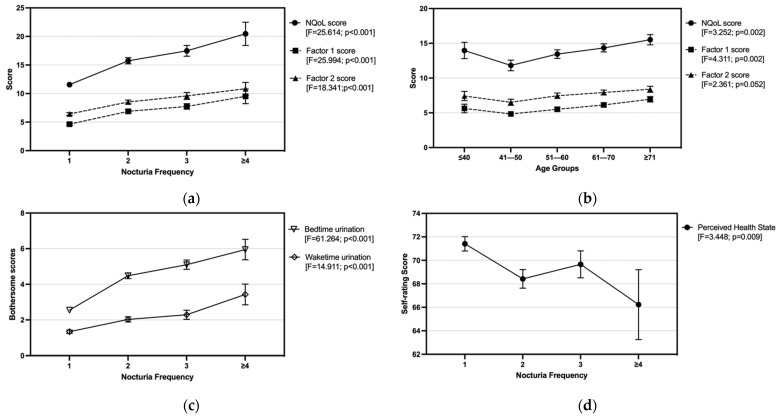
Nocturia-related QoL, bothersome caused by nocturia, and health state perceived by the individuals with nocturia, are presented as the trends of: (**a**) NQoL score and factor scores against the increasing nocturia frequency, and (**b**) against the advancing age groups, and (**c**) the bothersome score and (**d**) the health state score against the increasing nocturia frequency.

**Table 1 ijerph-18-09112-t001:** Characteristics of the studied populations, and comparisons between individuals with and without nocturia.

Variables	Total	Non-Nocturia	Nocturia	χ^2^
	*N* = 1239	*N* = 458	*N* = 781	*p*-Value
	Number (Percentage; Confidence Interval 95%)			
Demographics				
Mean age ± SD, years	56.57 ± 15.22	49.83 ± 14.84	60.52 ± 14.01	<0.001 ^#^
Married/cohabited	822 (66.3; 63.6–69.0)	306 (66.8; 62.3–71.1)	516 (66.1; 62.6–69.4)	0.789
Education < college	750 (60.5; 57.7–63.3)	217 (47.4; 42.7–52.1)	533 (68.2; 64.9–71.5)	<0.001
Current smoker	610 (49.2; 46.4–52.1)	202 (44.1; 40.7–47.7)	452 (57.9; 53.2–62.4)	<0.001
Daily drinker	312 (25.2; 22.8–27.7)	107 (23.4; 19.6–27.5)	205 (26.2; 23.2–29.5)	0.489
Waketime Urination				
Every hour	56 (4.5; 3.4–5.8)	13 (2.8; 1.5–4.8)	43 (5.5; 4.0–7.3)	<0.001
Every 2 h	330 (26.6; 24.2–29.2)	81 (17.7; 14.3–21.5)	249 (31.9; 28.6–35.3)	
Every 3 h	485 (39.1; 36.4–41.9)	173 (37.8; 33.3–42.4)	312 (40.0; 36.5–43.5)	
≥Every 4 h	368 (29.7; 27.2–32.3)	191 (41.7; 37.1–46.4)	177 (22.7; 19.8–25.8)	
Bothersome (0–10), mean ± SD	1.67 ± 2.20	0.93 ± 1.53	1.78 ± 2.27	<0.001 ^#^
Health & Illnesses				
Diabetes Mellitus	181 (14.6; 12.7–16.7)	34 (7.4; 5.2–10.2)	147 (18.8; 16.1–21.7)	<0.001
Hypertension	448 (36.2; 33.5–38.9)	98 (21.4; 17.7–25.4)	350 (44.8; 41.3–48.4)	<0.001
Diabetes Mellitus + Hypertension	153 (12.3; 10.6–14.3)	30 (6.6; 4.5–9.2)	123 (15.7; 13.3–18.5)	<0.001
Benign prostatic hyperplasia	96 (7.8; 6.3–9.4)	9 (2.0; 0.9–3.7)	87 (11.1; 0.9–13.6)	<0.001
Prostate cancer	5 (0.4; 0.1–0.9)	0 (0; 0–0.8)	5 (0.6; 0.2–1.5)	0.086
On Benign prostatic hyperplasia medication	65 (5.3; 431–6.6)	7 (1.5; 0.6–3.1)	58 (7.4; 5.7–9.5)	<0.001
Health state (0–100), mean ± SD	72.85 ± 12.57	76.79 ± 11.41	70.17 ± 12.64	<0.001 ^#^

^#^ Student’s *t*-test.

**Table 2 ijerph-18-09112-t002:** Distribution and age of respondents reporting different voiding episodes per bedtime.

Nocturia Episodes	Frequency	Percentage	Age (Mean ± SD, Years)
1	394	50.4	57.01 ± 13.53
2	254	32.5	61.06 ± 14.20
3	96	12.3	68.65 ± 9.11
≥4	37	4.7	73.00 ± 12.01

**Table 3 ijerph-18-09112-t003:** Correlational relationships between continuous variables.

Variables	NQoL Factor 2 Score	NQoL Factor 1 Score	NQoL Score	Nocturia Frequency	Waketime Urination Frequency
	Pearson Correlation Coefficient (r); *p*-Value				
Age	0.088; 0.014	0.137; <0.001	0.103; 0.004	0.329; <0.001	0.045; 0.206
Waketime urination frequency	−0.117; 0.001	−0.207; <0.001	−0.178; <0.001	−0.294; <0.001	
Nocturia frequency	0.253; <0.001	0.301; <0.001	0.294; <0.001		
NQoL score	0.954; <0.001	0.928; <0.001			
NQoL Factor 1 score	0.787; <0.001				

**Table 4 ijerph-18-09112-t004:** Chi-squared comparison of individual items of the NQoL scale between different nocturia frequency groups.

NQoL Items	Nocturia Frequency				χ^2^
	1 Time	2 Times	3 Times	≥4 Times	*p*-Value
	*N* = 391	*N* = 254	*N* = 96	*N* = 37	
	Number (Percentage) ^#^				
Rated the overall QoL poor/very poor	51 (13.0)	38 (15.0)	16 (16.7)	11 (29.7)	<0.001
Factor 1 (Sleep/Energy)					
Difficult to concentrate the next day	50 (12.8)	81 (31.9)	33 (34.4)	14 (37.8)	<0.001
Feel no energy the next day	73 (18.7)	103 (40.6)	51 (53.1)	18 (48.7)	<0.001
Required to nap at day time	92 (23.5)	124 (48.8)	59 (61.5)	17 (46.0)	<0.001
Lowered next day’s productivity	41 (10.5)	53 (20.9)	19 (19.8)	11 (29.7)	<0.001
Participate less in favorite activities	42 (10.7)	37 (14.6)	22 (22.9)	14 (37.8)	<0.001
Difficult to get enough sleep at night ^^^	121 (31.0)	123 (48.4)	61 (63.5)	19 (51.4)	<0.001
Factor 2 (Bother/Concern)					
Pay more attention of when and how much to drink	168 (43.0)	117 (46.1)	37 (38.5)	12 (32.4)	0.012
Difficult to get enough sleep at night ^^^	121 (31.0)	123 (48.4)	61 (63.5)	19 (51.4)	<0.001
Worried disturbing others in the house because of having to get up in the middle of night to urinate	81 (20.7)	75 (29.5)	28 (29.2)	16 (43.2)	0.007
Concerned that must get up in the middle of the night to urinate	85 (21.7)	99 (39.0)	55 (57.3)	20 (54.1)	<0.001
Worried that the condition of night urination will get worse	73 (18.7)	81 (31.9)	38 (39.6)	21 (56.8)	<0.001
Worried that there is no effective treatment for the condition of night urination	59 (15.1)	73 (28.7)	37 (38.5)	17 (46.0)	<0.001

^#^ Participants who have rated some of night’, ‘most of the night’ or ‘every night’ by participants. ^^^ Item repeated in both factor subscales.

## Data Availability

The raw data presented in this study are available on request from the corresponding author. The data are not publicly available due to research data governance policy of institution that has given ethical approval to this study.

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
