# Peer review of "A Comprehensive Community-Based Prevalence Study on Nocturia in Hong Kong Male Adults"

_ijerph, 2021, doi:10.3390/ijerph18179112_

Round 1

Author Response

As attached

Reviewer 2 Report

General evaluation

This paper addressed a very important topic: Prevalence of males with nocturia.

The aim of this study was to measure the nocturia prevalence across the full spectrum of male adults living in Hong Kong, where severity and associated quality-of-life (QoL) were also explored.

This study is important and has relevance for the general populations health, heath provider and for researchers. Following elements can be considered to strengthen the paper further:

Abstract:

OK

Background:

Nice introduction focusing on the definition of Nocturia and LUTS.

What is missing is the International classification of sleep disorders definition of Insomnia.

Insomnia is a patient-reported complaint of difficulty falling asleep or difficulty maintaining sleep, i.e., frequent awakenings, difficulty returning to sleep after awakenings, or awakening too early with inability to return to sleep. Although non-restorative sleep is often included as a symptom of insomnia, it has different epidemiological and functional correlates than other insomnia symptoms, including higher prevalence in young adults and a greater degree of daytime impairments such as sleepiness and fatigue. (Buysse, 2013)

The prevalence of insomnia disorder is approximately 10% to 20%, with approximately 50% having a chronic course.(Buysse, 2013)

Further, I miss the theoretical background and the Hypothesis behind the correlation Nocturia, sleep and QoL. In the publication of UM the correlation is with Insomnia or OSAS (Um et al., 2020).

Please add a theoretical construct and the hypotheses.

Methods:

Did you publish the translation process of the instrument?

Results:

Figure 2 is difficult to read, I suggest a normal table where always the Number of participants is visible.

Discussion

Please add a sentence on Figure 4d – interesting range. This shows multiple factors influencing perceived health status.

Nocturia is more prevalent in men with lower education. Please add a sentence that considers the self-care and the possibilities with a low income to buy health food, take care of the own body, do preventive tests…

Finally, I miss a sentence explaining the item "difficult to get enough sleep at night". Was this question correlated with nocturia? Many young men do not sleep enough. Many have problems to fall asleep and drink alcohol… I do not know what is the sleep behavior of young people in Hong Kong. I think that there could be a bad behavior of young men and that nocturia is one of a lot of other factors. Please add more knowledge from a sleep expertise view.

Buysse, D. J. (2013). Insomnia. JAMA, 309(7), 706. https://doi.org/10.1001/jama.2013.193

Um, Y. H., Oh, J.-H., Kim, T.-W., Seo, H.-J., Kim, S.-M., Chung, J.-S., Jeong, J.-H., & Hong, S.-C. (2020). Nocturia and Sleep: Focus on Common Comorbidities and Their Association with Obstructive Sleep Apnea. Sleep Medicine Research, 11(2), 59–64. https://doi.org/10.17241/smr.2020.00731

Author Response

As attached. 

Reviewer 3 Report

In this study, Yuen at al. investigated the prevalence of nocturia in Hong Kong male adults and it effects on their quality of life. While potentially interesting, there are several concerns that should be addressed before this study can be accepted for publishing.

  1. “Nocturia is one of the most popular lower urinary tract symptoms…”. Popular? Do you mean common?
  2. There are numerous typos, grammatical and sentence construction errors in this manuscript, which at times makes it very difficult to understand the intended meaning of some sentences (such as, for example “Numerous studies have reported prevalence of nocturia as both ≥1 and ≥2 voids per night, because the urology field believed clinical significance to affect energy metabolism and productivity is sought to at least two bedtime wakes”
  3. I am worried by the following statement “Implied consent of participation was assumed by completing the survey.” Explicit consent should and must have been received from the study participants.
  4. The authors have cited several studies reporting highly varied prevalence numbers of nocturia. After reading their manuscript, I am struggling to understand why their study is better than the previous ones.
  5. What is a definition of a frequent drinker in this study?
  6. “To our best knowledge, this is also the first study reporting” – such statement should be avoided
  7. Discussion is highly repetitive, for example “Many urologists argued that at least twice bed- time voids per night should be considered as clinically significant to affect energy metabolism and productivity [11-12]. Evidence has suggested that energy and productivity of any individuals could be substantially influenced if nocturia occurs at twice or more voiding episodes per night [12].” – these two sentences convey essentially the same information.
  8. 63% can hardly be called two thirds (as in the conclusion of this manuscript).

Author Response

As attached. 

Round 2

Reviewer 1 Report

I agree with the explanations and modifications made by the authors

Reviewer 3 Report

Most of the comments were appropriately addressed and I can now recommend this manuscript for publishing.